# Graph-level Representation Learning with Joint-Embedding Predictive Architectures

**Geri Skenderi**        *geri.skenderi@unibocconi.it*
*Bocconi Institute for Data Science and Analytics*
*Bocconi University*

**Hang Li**        *lihang4@msu.edu*
*Michigan State University*

**Jiliang Tang**        *tangjili@msu.edu*
*Michigan State University*

**Marco Cristani**        *marco.cristani@univr.it*
*University of Verona*

**Reviewed on OpenReview:** *https://openreview.net/forum?id=v47f4DwYZb*

## Abstract

Joint-Embedding Predictive Architectures (JEPAs) have recently emerged as a novel and powerful technique for self-supervised representation learning. They aim to learn an energy-based model by predicting the latent representation of a target signal $y$ from the latent representation of a context signal $x$. JEPAs bypass the need for negative and positive samples, traditionally required by contrastive learning while avoiding the overfitting issues associated with generative pretraining. In this paper, we show that graph-level representations can be effectively modeled using this paradigm by proposing a Graph Joint-Embedding Predictive Architecture (Graph-JEPA). In particular, we employ masked modeling and focus on predicting the *latent* representations of masked subgraphs starting from the latent representation of a context subgraph. To endow the representations with the implicit hierarchy that is often present in graph-level concepts, we devise an alternative prediction objective that consists of predicting the coordinates of the encoded subgraphs on the unit hyperbola in the 2D plane. Through multiple experimental evaluations, we show that Graph-JEPA can learn highly semantic and expressive representations, as shown by the downstream performance in graph classification, regression, and distinguishing non-isomorphic graphs. The code is available at `https://github.com/geriskenderi/graph-jepa`.

## 1 Introduction

Graph data is ubiquitous in the real world due to its ability to universally abstract various concepts and problems (Ma & Tang, 2021; Veličković, 2023). To deal with this widespread data structure, Graph Neural Networks (GNNs) (Scarselli et al., 2008; Kipf & Welling, 2016a; Gilmer et al., 2017; Veličković et al., 2017) have established themselves as a staple solution. Nevertheless, most applications of GNNs usually rely on ground-truth labels for training. The growing amount of graph data in fields such as bioinformatics, chemoinformatics, and social networks makes manual labeling laborious, sparking significant interest in unsupervised graph representation learning.

A particularly emergent area in this line of research is self-supervised learning (SSL). In SSL, alternative forms of supervision are created stemming from the input signal. This process is then typically followed by invariance-based or generative-based pretraining (Liu et al., 2023; Assran et al., 2023). Invariance-based approaches optimize the model to produce comparable embeddings for different views of the input signal. A

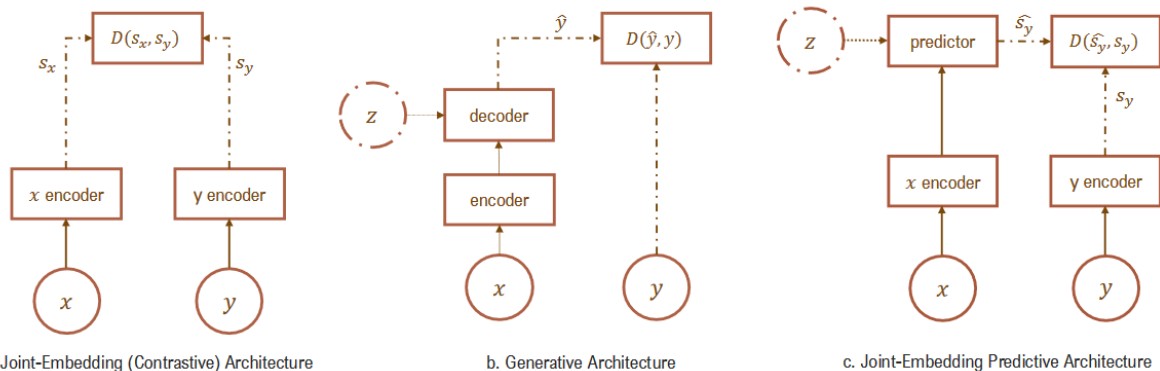

Figure 1: Illustration of the SSL approaches discussed in this paper: (a) Joint-Embedding (Contrastive) Architectures learn to create similar embeddings for inputs x and y that are compatible with each other and dissimilar embeddings otherwise. This compatibility is implemented in practice by creating different views of the input data. (b) Generative Architectures reconstruct a signal $y$ from an input signal $x$ by conditioning the decoder network on additional (potentially latent) variables $z$. (c) Joint-Embedding Predictive Architectures act as a bridge: They utilize a predictor network that processes the context $x$ and is conditioned on additional (potentially latent) variables to predict the embedding of the target $y$ *in latent space.*

common paradigm associated with this procedure is contrastive learning (Tian et al., 2020). Typically, these alternative views are created by a data augmentation procedure. The views are then passed through their respective encoder networks (which may share weights), as shown in Fig. 1a. Finally, an energy function, usually framed as a distance, acts on the latent embeddings. In the graph domain, several works have applied contrastive learning by designing graph-specific augmentations (You et al., 2020), using multi-view learning (Hassani & Khasahmadi, 2020) and even adversarial learning (Suresh et al., 2021). Invariance-based pretraining is effective but comes with several drawbacks i.e., the necessity to augment the data and process negative samples, which limits computational efficiency. In order to learn semantic embeddings that are useful for downstream tasks, the augmentations must also be non-trivial.

Generative-based pretraining methods, on the other hand, typically remove or corrupt portions of the input and predict them using an autoencoding procedure (Vincent et al., 2010; He et al., 2022) or rely on autoregressive modeling (Brown et al., 2020; Hu et al., 2020). Fig. 1b depicts the typical instantiation of these methods: The input signal $x$ is fed into an encoder network that constructs the latent representation, and from it a decoder generates $\hat{y}$, the data corresponding to the target signal $y$. The energy function is then applied in data space, often through a reconstruction error. Generative models generally display strong overfitting tendencies (van den Burg & Williams, 2021) and can be non-trivial to train due to issues such as mode collapse (Adiga et al., 2018). Moreover, the features they learn are not always useful for downstream tasks (Meng et al., 2017). An intuitive explanation of this problem is that generative models have to estimate a data distribution that is usually quite complex, so the latent representations must be directly descriptive of the whole data space (Loaiza-Ganem et al., 2022). This can become even more problematic for graphs because they live in a non-Euclidean and inhomogenous data space. Despite the aforementioned issues, masked autoencoding has recently shown promising results also in the graph domain with appropriately designed models (Hou et al., 2022; Tan et al., 2023).

Inspired by the innovative Joint-Embedding Predictive Architecture (JEPA) (LeCun, 2022; Assran et al., 2023), we propose Graph-JEPA, a JEPA for the graph domain that can learn graph-level representations by bridging contrastive and generative models. As illustrated in Fig. 1c, a JEPA has two encoder networks that receive the input signals and produce the corresponding representations. The two encoders can potentially be different models and don't need to share weights. A predictor module outputs a prediction of the latent representation of one signal based on the other, possibly conditioned on another variable. Graph-JEPA does not require any negative samples or complex data augmentation, and by operating in the latent space avoids the pitfalls associated with learning high-level details needed to fit the data distribution. However,

the graph domain presents several challenges needed to properly design such an architecture: Context and target extraction, designing a latent prediction task optimal for graph-level concepts, and learning expressive representations. In response to these questions, we equip Graph-JEPA with a specific masked modeling objective. The input graph is first divided into several subgraphs, and then the latent representation of randomly chosen target subgraphs is predicted given a context subgraph. The subgraph representations are consequently pooled to create a graph-level representation that can be used for downstream tasks.

The nature of graph-level concepts is often assumed to be hierarchical (Ying et al., 2018). We conjecture that the typical latent reconstruction objective used in current JEPA formulations is not enough to provide optimal downstream performance. To this end, we design a prediction objective that starts by expressing the target subgraph encoding as a high-dimensional description of the hyperbolic angle. The predictor module is then tasked with predicting the location of the target in the 2D unit hyperbola. This prediction is compared with the target coordinates obtained using the aforementioned hyperbolic angle. In this self-predictive setting, we explain why the stop-gradient operation and a simple predictor parametrization are useful to prevent representation collapse. To experimentally validate our approach, we evaluate Graph-JEPA against established contrastive and generative graph-level SSL methods across various graph datasets from different domains. Our proposed method demonstrates superior performance, outperforming most competitors while maintaining efficiency and ease of training. Notably, we observe from our experiments that Graph-JEPA can run up to 2.5x faster than Graph-MAE (Hou et al., 2022) and 8x faster than MVGRL (Hassani & Khasahmadi, 2020). Finally, we empirically demonstrate Graph-JEPA's ability to learn highly expressive graph representations by showing that a linear classifier trained on the learned representations almost perfectly distinguishes pairs of non-isomorphic graphs that the 1-WL test cannot differentiate.

## 2 Related work

### 2.1 Self-Supervised Graph Representation Learning

Graph Neural Networks (Wu et al., 2019; Scarselli et al., 2008; Kipf & Welling, 2016a; Veličković et al., 2017) are now established solutions to different graph machine learning problems such as node classification, link prediction, and graph classification. Nevertheless, the cost of labeling graph data is relatively high, given the immense variability of graph types and the information they can represent. To alleviate this problem, SSL on graphs has become a research frontier, where we can distinguish between two major groups of methods(Xie et al., 2022b; Liu et al., 2023):

**Contrastive Methods.** Contrastive learning methods usually minimize an energy function (Hinton, 2002; Gutmann & Hyvärinen, 2010) between different views of the same data. InfoGraph (Sun et al., 2019) maximizes the mutual information between the graph-level representation and the representations of substructures at different scales. GraphCL (You et al., 2020) works similarly to distance-based contrastive methods in the imaging domain. The authors first propose four types of graph augmentations and then perform contrastive learning based on them. The work of (Hassani & Khasahmadi, 2020) goes one step further by contrasting structural views of graphs. They also show that a large number of views or multiscale training does not seem to be beneficial, contrary to the image domain. Another popular research direction for contrastive methods is learning graph augmentations and how to leverage them efficiently (Suresh et al., 2021; Jin et al., 2021). Contrastive learning methods typically require a lot of memory due to data augmentation and negative samples. Graph-JEPA is much more efficient than typical formulations of these architectures, given that it does not require any augmentations or negative samples. Another major difference is that the prediction in latent space in JEPAs is done through a separate predictor network rather than using the common Siamese structure (Bromley et al., 1993)(Fig. 1a vs. 1c).

**Generative Methods.** The goal of generative models is to recover the data distribution, an objective that is typically implemented through a reconstruction process. In the graph domain, most generative architectures that are also used for SSL are extensions of Auto-Encoder (AE) models (Hinton & Zemel, 1993) architectures. These models learn an embedding from the input data and then use a reconstruction objective with (optional) regularization to learn the data distribution. Kipf & Welling (2016b) extended

the framework of different AEs and Variational AEs (Kingma & Welling, 2013) to graphs by using a GNN as an encoder and the reconstruction of the adjacency matrix as the training objective. However, the results on downstream tasks with embeddings learned in this way are often unsatisfactory compared with contrastive learning methods, a tendency also observed in other domains (Liu et al., 2023). A recent and promising direction is masked autoencoding (MAE) (He et al., 2022), which has proved to be a very successful framework for the image and text domains. GraphMAE (Hou et al., 2022) is an instantiation of MAEs in the graph domain, where the node attributes are perturbed and then reconstructed, providing a paradigm shift from the structure learning objective of GAEs. S2GAE (Tan et al., 2023) is one of the latest GAEs, which focuses on reconstructing the topological structure but adds several auxiliary objectives and additional designs. Our architecture differs from generative models in that it learns to predict directly in the latent space, thereby bypassing the necessity of remembering and overfitting high-level details that help maximize the data evidence (Fig. 1b vs. 1c).

## 2.2 Joint-Embedding Predictive Architectures

Joint-Embedding Predictive Architectures (LeCun, 2022) are a recently proposed design for SSL. The idea is similar to both generative and contrastive approaches, yet JEPAs are non-generative since they cannot directly predict $y$ from $x$, as shown in Fig. 1c. The energy of a JEPA is given by the prediction error in the embedding space, not the input space. These models can intuitively be understood as a way to capture abstract dependencies between $x$ and $y$, potentially given another latent variable $z$. It is worth noting that the different models comprising the architecture may differ in terms of structure and parameters. An in-depth explanation of Joint-Embedding Predictive Architectures and their connections to human representation learning is provided by LeCun (2022). Some works acknowledged that latent self-predictive architectures were effective (Grill et al., 2020; Chen & He, 2021) even before JEPAs effectively became synonymous with this concept. Inspired by these trends, a number of related works have tried to employ latent prediction objectives for graph SSL, showing advantages mostly compared to contrastive learning. Thakoor et al. (2021) perform latent self-prediction on augmented views of a graph in a similar fashion to BYOL (Grill et al., 2020), while Zhang et al. (2021) rely on ideas from Canonical Correlation Analysis to frame a learning objective that preserves feature invariance and forces decorrelation when necessary. The work of Lee et al. (2022) presents a model that learns latent positive examples through a k-NN and clustering procedure in the transductive setting, while Xie et al. (2022a) combine instance-level reconstruction (generative pretraining) and feature-level invariance (latent prediction). Given that these models learn using a latent self-predictive objective, similar to ours, we will refer to them also using the term self-predictive in the rest of the paper. Unlike previously proposed methods, Graph-JEPA operates exclusively in the latent space and implements a novel training task without the need for data augmentation. At the current state of the art, the JEPA framework has been implemented for images (Assran et al., 2023), video (Bardes et al., 2023b;a), and audio (Fei et al., 2023). We propose the first architecture implementing the modern JEPA principles for the graph domain and use it to learn graph-level representations.

## 3 Method

**Notation and General Overview.** We consider graphs $G$ defined as $G = (V, E)$ where $V = \{v_1 \dots v_N\}$ is the set of nodes, with a cardinality $|V| = N$, and $E = \{e_1 \dots e_M\}$ is the set of edges, with a cardinality $|E| = M$. For simplicity of the exposition, we consider symmetric, unweighted graphs, although our method can be generalized to weighted or directed graphs. In this setting, $G$ can be represented by an adjacency matrix $A \in \{0,1\}^{N \times N}$, with $A_{ij} = 1$ if nodes $v_i$ and $v_j$ are connected and 0 otherwise. Finally, let the neighborhood of a node $v$ be defined as $\mathcal{N}(v) = \{u \mid (u, v) \in E\}$. Fig. 2 provides the overview of the proposed architecture. The high-level idea of Graph-JEPA is to divide the input graph into subgraphs (patches) (He et al., 2023) and then predict the representation of a randomly chosen target subgraph from the representation of a single context subgraph (Assran et al., 2023). Again, we would like to stress that this masked modeling objective is realized in latent space without the need for negative samples. The subgraph representations are then pooled to create a vector representation for the whole graph, i.e., a graph-level representation. Therefore, the learning procedure consists of a sequence of operations: i) Spatial Partitioning; ii) Subgraph Embedding; iii) Context and Target Encoding; iv) Latent Target Prediction.

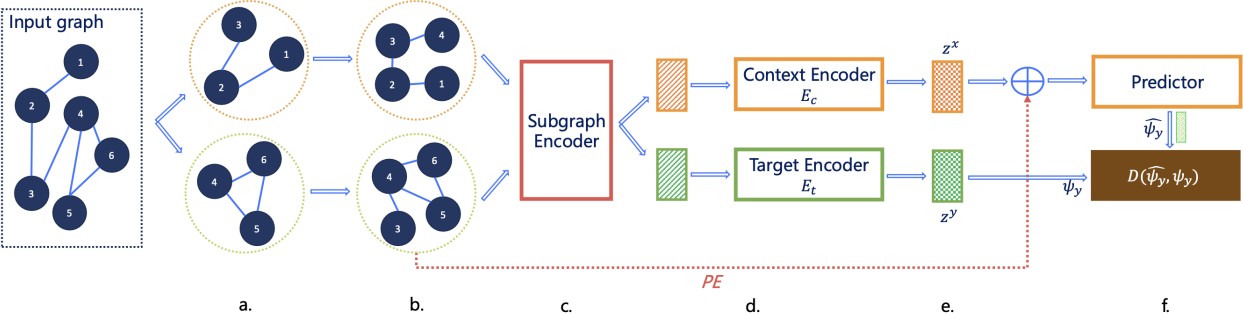

Figure 2: A complete overview of Graph-JEPA. We first extract non-overlapping subgraphs (patches) (a.), perform a 1-hop neighborhood expansion (b.), and encode the subgraphs with a GNN to learn feature vectors for the context and target (c.). Afterward, the context and target feature vectors are fed into their respective encoders (d.). The embeddings generated from the target encoder produce the target subgraphs hyperbolic coordinates $\psi_y$. On the other hand, the encoded context is fed into a predictor network, which is also conditioned on the positional embedding of the target subgraph, to then predict the coordinates $\hat{\psi}_y$ for the target subgraph (e.). A regression loss $D$, based on the distance in latent space, acts as the learning objective (f.). Note that the extracted subgraphs in (a.) and (b.) are meant for illustrative purposes only, as in practice, we use multiple target subgraphs for a given context. Furthermore, the number of nodes in each subgraph can vary.

## 3.1 Spatial Partitioning

We base the initial part of our architecture on the recent work of (He et al., 2023), but similar ideas have been proposed before for Graph SSL (Jin et al., 2020). This step consists of creating different subgraphs (patches), similar to how Vision Transformers (ViT) (Dosovitskiy et al., 2020) operate on images. Supported by experimental evidence in other data modalities (Assran et al., 2023; Fei et al., 2023), we conjecture that a critical aspect of training JEPAs is building a self-predictive task that is non-trivial and aligned with the characteristics of downstream problems. Therefore, we proceed by providing as input a graph partitioning, i.e., a collection of $p$ subsets, $\{V_1, \ldots, V_p\}$ such that $V_i \cap V_j = \emptyset$ for $i \neq j$, $|V_i| = N/k$, and $\bigcup_i V_i = V$ (Fig. 2a). The rationale behind this choice is that the partition will inherently embed a notion of locality and spatial awareness into the learned representation via the latent self-predictive task, which is beneficial for downstream performance. In practice, we rely on the METIS (Karypis & Kumar, 1998) graph clustering algorithm, which partitions the graph such that the number of intra-cluster links is much higher than inter-cluster links, i.e., the number of edges whose incident vertices belong to different partitions is minimized. METIS is a multilevel algorithm, meaning that it first coarsens the graph, then performs partitioning, and finally, during an uncoarsening phase, projects the partitions back to the original graph, with their boundaries refined to reduce edge cuts and enhance intra-cluster density. This way, we can adequately embed local properties and community structure in the model input. It becomes pretty direct to see how these properties, especially the multilevel nature of the partitioning, relate to the hierarchical nature of graph-level concepts.

Note that having non-overlapping subgraphs can be problematic in practice since edges can be lost in this procedure, and it is possible to end up with edgeless subgraphs. Furthermore, signal propagation over the subgraphs becomes impossible due to the partitioning. Therefore, we implement a 1-hop neighborhood expansion (Fig. 2b), where each partition is transformed into $G'_i = (V'_i, E'_i)$, such that $V'_i = V_i \cap \mathcal{N}(V_j), \forall j \neq i$ and $E'_i = \{(u, v) \mid u, v \in V'_i \text{ and } (u, v) \in E\}$. Simply put, all corresponding neighboring nodes from the original input graph are connected with each node in the subgraph. The model can then aggregate information from neighboring regions by incorporating these overlaps when learning subgraph embeddings. This enables it to capture broader graph-level patterns while retaining the properties of working with smaller, localized subgraphs. Such a property is crucial for the next step of Graph-JEPA, described in the following section.

## 3.2 Subgraph Embedding

After partitioning the graph, we learn a representation for each subgraph through a GNN (Fig. 2c.). The specific choice of the GNN is arbitrary and depends on what properties one wishes to induce in the representation. The learned node embeddings are mean pooled to create a vector representation for each subgraph: $\{h_1...h_p\}, h \in \mathbb{R}^d$. These embeddings will be used as context or target variables in the JEPA framework (Fig. 1c.). Using only these embeddings can render the latent self-predictive too difficult, therefore, we include positional information for each subgraph. The positional embedding is implemented as the maximum Random Walk Structural Embedding (RWSE) of all the nodes in that subgraph. In this way, each patch's position is characterized consistently and globally. Formally, a RWSE (Dwivedi et al., 2021) for a node $v$ can be defined as:

$$P_v = (M_{ii}, M_{ii}^2, \ldots, M_{ii}^k), \tag{1}$$

where $P_v \in \mathbb{R}^k$, $M^k = (D^{-1}A)^k$ is the random-walk transition matrix of order $k$, $D$ is a diagonal matrix containing the degrees of each node, and $i$ is the index $v$ in $A$. $M_{ii}^k$ encodes the probability of node $v$ landing to itself after a $k$-step random walk. It is noteworthy to see that such an embedding defines a signal over the graph, if we view the positional embedding as a function $\phi : V \to \mathbb{R}^k$. Furthermore, under a few reasonable assumptions, i.e., having connected, non-bipartite subgraphs with different connectomes, this function is injective $\phi(u) = \phi(v) \Rightarrow u = v, \forall u, v \in V$. This property follows directly from the definition of $M^k$ and the stationary distribution of a graph random walk Lovász (1993). Given a subgraph $l$, we define its RWSE as:

$$P_l = \max_{v \in V_l} P_v, \tag{2}$$

where the max operation is performed elementwise (on the nodes). From the above, we define the positional information of each subgraph as being essentially contained in the node with the highest degree[1], which will act as an anchor reference when predicting the target subgraphs' latent representations. Therefore, our choice of positional embedding will provide a global context for the predictor network, as the positional embeddings of these anchor nodes can identify the location of their respective subgraphs. Intuitively, knowing $P_l$ is useful for the prediction task because the features present in the most representative node will have been diffused the most (by the GNN) into the subgraph.

## 3.3 Context and Target Encoding

Given the subgraph representations and their respective positional embeddings, we frame the Graph-JEPA prediction task in a similar manner to I-JEPA (Assran et al., 2023). The goal of the network is to predict the latent embeddings of randomly chosen target subgraphs, given one random context subgraph, conditioned on the positional information of each target subgraph. At each training step, we choose one random subgraph as the context $x$ and $m$ others as targets $Y = \{y_1, \ldots, y_m\}$. Using a single context is arbitrary but provides numerous advantages. Firstly, it guarantees that we can work with small and large graphs, as shown in Sec. 4.2. Secondly, it implicitly regularizes the model in learning long-range dependencies by predicting targets that are far (in terms of shortest path distance) from the context. We empirically validate this second aspect by testing for graph isomorphisms in Sec. 4.2. These subgraphs are processed by the respective context and target encoders (Fig. 2d), which are parametrized by Transformer encoder blocks (without self-attention for the context) where normalization is applied at first (Xiong et al., 2020). The target encoder uses Hadamard self-attention (He et al., 2023), but other choices, such as the standard self-attention mechanism (Vaswani et al., 2017), are perfectly viable. We can summarize this step as:

$$z^x = E_c(x), \ Z^y = E_t(Y), \tag{3}$$

with $z^x \in \mathbb{R}^d$ and $Z^y \in \mathbb{R}^{m \times d}$. At this stage, we can use the predictor network to directly predict $Z^y$ from $z^x$. This is the typical formulation of JEPAs, followed by the popular work of Assran et al. (2023). We argue that learning to organize concepts for abstract objects such as graphs or networks directly in Euclidean space is suboptimal. In the following subsection, we propose a simple trick to bypass this problem using the encoding and prediction mechanisms in Graph-JEPA. A discussion in Sec. 4.3 will provide additional insights.

---

[1]The more suitable term would be in-degree, but there is no difference in the undirected case.

### 3.4 Latent Target Prediction

Learning hierarchically consistent concepts (Deco et al., 2021) is considered a crucial aspect of human learning, especially during infancy and young age (Rosenberg & Feigenson, 2013). Networks in the real world often conform to some concept of hierarchy (Moutsinas et al., 2021), and this assumption is frequently used when learning graph-level representations (Ying et al., 2018). Thus, we conjecture that Graph-JEPA should operate in a hyperbolic space, where learned embeddings implicitly organize hierarchical concepts (Nickel & Kiela, 2017; Zhao et al., 2023). This gives rise to another issue: commonly used hyperbolic (Poincaré) embeddings are known to have several tradeoffs between dimensionality and performance (Sala et al., 2018; Guo et al., 2022), which severely limits the expressive ability of the model. Given that graphs can describe very abstract concepts, high expressivity in terms of model parameters is preferred. In simple words, we would ideally like to have a high-dimensional latent code that has a concept of hyperbolicity built into it.

To achieve this, we think of the target embedding as a high-dimensional representation of the hyperbolic angle, which allows us to describe each target patch through its position in the 2D unit hyperbola. Formally, given a target patch $l$, its embedding $Z_l^y$ and positional encoding $P_l$, we express the latent target as:

$$\psi_l^y = \begin{pmatrix} cosh(\alpha_l^y) \\ sinh(\alpha_l^y) \end{pmatrix}, \ \alpha_l^y = \frac{1}{N} \sum_{n=1}^{d} Z_l^{y(n)}, \tag{4}$$

where $cosh(\cdot)$ and $sinh(\cdot)$ are the hyperbolic cosine and sine functions respectively. The predictor module is then tasked with directly locating the target in the unit hyperbola, given the context embedding and the target patch's positional encoding Fig 2e):

$$\hat{\psi}_l^y = W_2(\sigma(W_1(z^x + P_l) + b_1)) + b_2, \tag{5}$$

where $W_n$ and $b_n$ represent the n-th weight matrix and bias vector (i.e., n-th fully connected layer), $\sigma$ is an elementwise non-linear activation function, and $\hat{\psi}_l^y \in \mathbb{R}^2$. This allows us to frame the learning procedure as a low-dimensional regression problem, and the whole network can be trained end-to-end (Fig. 2f). In practice, we use the smooth L1 loss as the distance function, as it is less sensitive to outliers compared to the typical L2 loss (Girshick, 2015):

$$L(y, \hat{y}) = \frac{1}{N} \sum_{n=1}^{N} s_n, \quad s_n = \begin{cases} 0.5(y_n - \hat{y}_n)^2/\beta, & \text{if } |y - \hat{y}| < \beta \\ |y - \hat{y}| - 0.5\beta, & \text{otherwise} \end{cases} \tag{6}$$

Thus, we are effectively measuring how far away the context and target patches are in the unit hyperbola of the plane by first describing the subgraphs through a high dimensional latent code (Eq. 4). This representation balances the expressivity of high-dimensional embeddings and the hierarchical organization provided by hyperbolic geometry. Intuitively, the hyperbolic angle $\alpha$ represents an aggregate measure of the subgraph's latent code, compressed into a scalar value. In high dimensions, this scalar value will concentrate around one value, which provides the base level of the hierarchy. At the same time, the encoder networks have the freedom to produce largely deviant latent codes that lead to higher hierarchy levels. A visualization of this process is provided in the Appendix (Fig. 4). We explicitly show the differences between this choice and using the Euclidean or (Poincaré) Hyperbolic distance as energy functions for the training procedure in Sec. 4.3. Our proposed pretraining objective forces the context encoder to understand the differences in the hyperbolic angle between the target patches, which can be thought of as establishing an implicit hierarchy between them.

**Preventing Representation Collapse.** JEPAs are based on a self-distillation procedure. Therefore, they are by definition susceptible to representation collapse (Assran et al., 2023). This is due to the nature of the learning process, where both the context and target representations have to be learned. We formalize this intuition with an example and argue why there is a need to adopt two well-known training tricks

that are prevalent in the literature to prevent representation collapse: i) The stop-gradient operation on the target encoder followed by a momentum update (using an Exponential Moving Average (EMA) of the context encoder weights) (Grill et al., 2020; Chen & He, 2021); ii) a simpler parametrization of the predictor compared to the context and target networks (in terms of parameters)(Chen et al., 2020; Baevski et al., 2022). Let us simplify the problem through the following assumptions: i) The predictor network is linear; ii) There is a one-to-one correspondence between context and target patches (this holds in practice due to Eq. 5); iii) The self-predictive task is a least-squares problem in a finite-dimensional vector space. Based on these assumptions, we can rewrite the context features as $X \in \mathbb{R}^{n \times d}$, the target coordinates as $Y \in \mathbb{R}^{n \times 2}$, and the weights of the linear model as $W \in \mathbb{R}^{d \times 2}$. The optimal weights of the predictor are given by solving:

$$\arg\min_W \|XW - Y\|^2, \tag{7}$$

where $\|.\|$ indicates the Frobenius norm. The (multivariate) OLS estimator can give the solution to this problem by setting $W$ to:

$$W = (X^T X)^{-1} X^T Y. \tag{8}$$

Plugging Eq. 8 into Eq. 7 and factorizing $Y$, the least squares solution leads to the error:

$$\left\| (X(X^T X)^{-1} X^T - I_n)Y \right\|^2. \tag{9}$$

Thus, the optimality of a linear predictor is defined by the orthogonal projection of $Y$ onto the orthogonal complement of a subspace of $Col(X)$. As is commonly understood, this translates to finding the linear combination of $X$ that is closest, in terms of $\|\cdot\|^2$, to $Y$. Similarly to what was shown by Richemond et al. (2023), we argue that this behavior unveils a key intuition: The target encoder, *which estimates $Y$* must not share weights or be optimized via the same optimizer as the context encoder. If that were the case, the easiest solution to Eq. 9 would be learning a representation that is orthogonal to itself, i.e., the **0** vector, leading to representation collapse. Using a well-parametrized EMA update is what allows us to bypass this problem. In practice, even with the slower dynamics induced by the EMA procedure, it is possible to immediately encounter a degenerate solution with a non-linear and highly expressive network. For instance, consider a scenario where the target subgraphs are straightforward and similar. In this case, if the predictor network is sufficiently powerful, it can predict the target representations even without a well-learned context representation. Since the target encoder weights are updated via the EMA procedure, the learned representations will tend to be uninformative. Therefore, implementing the predictor network as a simpler and less expressive network is crucial to achieving the desired training dynamics.

## 4 Experiments

The experimental section introduces the empirical evaluation of the Graph-JEPA model in terms of downstream performance on different graph datasets and tasks, along with additional studies on the latent space's structure and the encoders' parametrization. Furthermore, a series of ablation studies are presented in order to elucidate the design choices behind Graph-JEPA.

### 4.1 Experimental Setting

We use the TUD datasets (Morris et al., 2020) as commonly done for graph-level SSL (Suresh et al., 2021; Tan et al., 2023). We utilize seven different graph-classification datasets: PROTEINS, MUTAG, DD, REDDIT-BINARY, REDDIT-MULTI-5K, IMDB-BINARY, and IMDB-MULTI. We report the accuracy of ten-fold cross-validation for all classification experiments over five runs (with different seeds). It is worth noting that we retrain the Graph-JEPA model for each fold without ever having access to the testing partition in both the pretraining and fine-tuning stages. We use the ZINC dataset for graph regression and report the Mean Squared Error (MSE) over ten runs (with different seeds), given that the testing partition is already separated. To produce the unique graph-level representations, we feed all the subgraphs through the trained target encoder and then use mean pooling, obtaining a single feature vector $z_G \in \mathbb{R}^d$ that represents the whole graph. This vector representation is then used to fit a *linear model* with L2 regularization for the downstream task. Specifically, we employ Logistic Regression with L2 regularization on the classification

datasets and Ridge Regression for the ZINC dataset. For the datasets that do not natively have node and edge features, we use a simple constant (0) initialization. The subgraph embedding GNN (Fig. 2c.) consists of the GIN operator with support for edge features (Hu et al., 2019), often referred to as GINE. The neural network modules were trained using the Adam optimizer (Kingma & Ba, 2014) and implemented using PyTorch (Paszke et al., 2019) and PyTorch-Geometric (Fey & Lenssen, 2019), while the linear classifiers and cross-validation procedure were implemented through the Scikit-Learn library (Pedregosa et al., 2011). All experiments were performed on Nvidia RTX 3090 GPUs. Tables 7 and 8 in the Appendix contain the dataset statistics and the JEPA-specific hyperparameters used in the following experiments, respectively.

## 4.2 Downstream Performance

For the experiments on downstream performance, we follow Suresh et al. (2021) and also report the results of a fully supervised Graph Isomorphism Network (GIN) (Xu et al., 2018), denoted F-GIN in Tab. 1. We compare Graph-JEPA against four contrastive methods, two generative methods, and one latent self-predictive method (Xie et al., 2022a) (which also regularizes through instance-level reconstruction). As can be seen in Tab. 1, Graph-JEPA achieves competitive results on all datasets, setting the state-of-the-art as a pretrained backbone on five different datasets and coming second on one (out of eight total). Overall, our proposed framework learns semantic embeddings that work well on different graphs, showing that Graph-JEPA can be utilized as a general pretraining method for graph-level SSL. Notably, Graph-JEPA works well for both classification and regression and performs better than a supervised GIN on all classification datasets. We also provide results with BGRL (Thakoor et al., 2021), a node-level latent self-predictive strategy. We train this model using the official code and hyperparameters and then mean-pool the node representations for the downstream task. The results are underwhelming compared to the models reporting graph-level performance, which is to be expected considering that methods that also perform well on graph-level learning are appropriately designed.

One notable aspect of the downstream performance of Graph-JEPA is the variance in the results. We conjecture that this variance in the results is due to three main factors. First, the randomness inherent in the subgraph partitioning process can lead to the selection of trivial context subgraphs, making it difficult to accurately predict latent representations of the target subgraphs. Second, while the JEPA paradigm has proven effective in the graph domain, it is susceptible to representation collapse, mainly when the previously mentioned issue is combined with positional embeddings of smaller, similar subgraphs. This would lead to nearly identical latent representations and, thus, reduced downstream performance. The issue is also compounded due to the fact that we did not extensively tune hyperparameters, as our focus was on demonstrating the architecture's efficacy with a simple training regimen. Lastly, in contrast to standard protocols that involve training on the entire dataset before fine-tuning, we trained the self-supervised model exclusively on the training set to ensure fairness. Despite this, average-case performance remains the most crucial aspect of our experimental validation.

We further explore the performance of our model on the synthetic EXP dataset (Abboud et al., 2020), compared to end-to-end supervised models. This experiment aims to empirically verify if Graph-JEPA can learn highly expressive graph representations (in terms of the commonly used WL hierarchy (Morris et al., 2019)) without relying on supervision. The results in Tab. 2 show that our model is able to perform much better than commonly used GNNs. Given its local and global exchange of information, this result is to be expected. Most importantly, Graph-JEPA almost matches the flawless performance achieved by He et al. (2023), who train fully supervised.

## 4.3 Exploring the Graph-JEPA Latent Space

As discussed in Sec. 3.4, the choice of energy function has a big impact on the learned representations. Given the latent prediction task of Graph-JEPA, we expect the latent representations to display hyperbolicity. The predictor network is linearly approximating the behavior of the unit hyperbola such that it best matches the generated target coordinates (Eq. 6). Thus, the network is actually trying to estimate a space that can be considered a particular section of the hyperboloid model (Reynolds, 1993), where hyperbolas appear as geodesics. We are, therefore, evaluating our energy in a restricted part of hyperbolic space. As mentioned

Table 1: Performance of different graph SSL techniques on various TUD benchmark datasets, ordered by pretraining type: contrastive, generative, and self-predictive. F-GIN is an end-to-end supervised GIN and serves as a reference for the performance values. The results of the competitors are taken as the best values from (Hassani & Khasahmadi, 2020; Suresh et al., 2021; Tan et al., 2023). "-" indicates missing values from the literature. The **best results** are reported in **boldface**, and the second best are underlined. For the sake of completeness, we also report the training and evaluation results of GraphMAE on the DD, REDDIT-M5, and ZINC datasets in *italics*, along with the results of a node-level self-predictive method (BGRL), which does not originally report results on graph-level tasks.

| Model | PROTEINS ↑ | MUTAG ↑ | DD ↑ | REDDIT-B ↑ | REDDIT-M5 ↑ | IMDB-B ↑ | IMDB-M ↑ | ZINC ↓ |
|---|---|---|---|---|---|---|---|---|
| F-GIN | 72.39 ± 2.76 | 90.41 ± 4.61 | 74.87 ± 3.56 | 86.79 ± 2.04 | 53.28 ± 3.17 | 71.83 ± 1.93 | 48.46 ± 2.31 | 0.254 ± 0.005 |
| InfoGraph (Sun et al., 2019) | 72.57 ± 0.65 | 87.71 ± 1.77 | 75.23 ± 0.39 | 78.79 ± 2.14 | 51.11 ± 0.55 | 71.11 ± 0.88 | 48.66 ± 0.67 | 0.890 ± 0.017 |
| GraphCL (You et al., 2020) | 72.86 ± 1.01 | 88.29 ± 1.31 | 74.70 ± 0.70 | 82.63 ± 0.99 | 53.05 ± 0.40 | 70.80 ± 0.77 | 48.49 ± 0.63 | 0.627 ± 0.013 |
| MVGRL (Hassani & Khasahmadi, 2020) | - | - | - | 84.5 ± 0.6 | - | 74.2 ± 0.7 | 51.2 ± 0.5 | - |
| AD-GCL-FIX (Suresh et al., 2021) | 73.59 ± 0.65 | 89.25 ± 1.45 | 74.49 ± 0.52 | 85.52 ± 0.79 | 53.00 ± 0.82 | 71.57 ± 1.01 | 49.04 ± 0.53 | 0.578 ± 0.012 |
| AD-GCL-OPT (Suresh et al., 2021) | 73.81 ± 0.46 | 89.70 ± 1.03 | 75.10 ± 0.39 | 85.52 ± 0.79 | 54.93 ± 0.43 | 72.33 ± 0.56 | 49.89 ± 0.66 | 0.544 ± 0.004 |
| GraphMAE (Hou et al., 2022) | 75.30 ± 0.39 | 88.19 ± 1.26 | *74.27 ± 1.07* | 88.01 ± 0.19 | *46.06 ± 3.44* | 75.52 ± 0.66 | 51.63 ± 0.52 | *0.935 ± 0.034* |
| S2GAE (Tan et al., 2023) | **76.37 ± 0.43** | 88.26 ± 0.76 | - | 87.83 ± 0.27 | - | **75.76 ± 0.62** | **51.79 ± 0.36** | - |
| BGRL (Thakoor et al., 2021) | 70.99 ± 3.86 | 74.99 ± 8.83 | 71.52 ± 2.97 | 50 ± 0 | 20 ± 0.1 | 0.5 ± 0 | 0.33 ± 0 | 1.2 ± 0.011 |
| LaGraph (Xie et al., 2022a) | 75.2 ± 0.4 | 90.2 ± 1.1 | 78.1 ± 0.4 | 90.4 ± 0.8 | 56.4 ± 0.4 | 73.7 ± 0.9 | - | - |
| Graph-JEPA | 75.68 ± 3.78 | **91.25 ± 5.75** | **78.64 ± 2.35** | **91.99 ± 1.59** | **56.73 ± 1.96** | 73.68 ± 3.24 | 50.69 ± 2.91 | **0.434 ± 0.014** |

Table 2: Classification accuracy on the synthetic EXP dataset (Abboud et al., 2020), which contains 600 pairs of non-isomorphic graphs that are indistinguishable by the 1-WL test. Note that the competitor models are all trained with *end-to-end supervision*. The **best result** is reported in boldface, and the second best is underlined. Performances for all competitor models are taken from (He et al., 2023).

| Model | Accuracy ↑ |
|---|---|
| GCN (Kipf & Welling, 2016a) | 51.90 ± 1.96 |
| GatedGCN (Bresson & Laurent, 2017) | 51.73 ± 1.65 |
| GINE (Xu et al., 2018) | 50.69 ± 1.39 |
| GraphTransformer (Dwivedi & Bresson, 2020) | 52.35 ± 2.32 |
| Graph-MLP-Mixer (He et al., 2023) | **100.00 ± 0.00** |
| *Graph-JEPA* | 98.77 ± 0.99 |

before, we find this task to offer great flexibility as it is straightforward to implement and it is computationally efficient compared to the hyperbolic distance used to typically learn hyperbolic embeddings in the Poincaré ball model (Nickel & Kiela, 2017). Tab. 3 provides empirical evidence for our conjectures regarding the suboptimality of Euclidean or Poincaré embeddings on 4 out of the 8 datasets initially presented in Tab. 1, where we make sure to choose different graph types for a fair comparison. The results reveal that learning the distance between patches in the 2D unit hyperbola provides a simple way to get the advantages of both embedding types. Hyperbolic embeddings must be learned in lower dimensions due to stability issues (Yu & De Sa, 2021), while Euclidean ones do not properly reflect the dependencies between subgraphs and the hierarchical nature of graph-level concepts. Our results suggest that the hyperbolic (Poincaré) distance is generally a better choice than the Euclidean distance in lower dimensions, but it is computationally unstable and expensive in high dimensions. The proposed approach provides the best overall results. We provide a qualitative example of how the embedding space is altered from our latent prediction objective in Fig. 3.

## 4.4 Additional Insights and Ablation Studies

**Model Efficiency.** In an attempt to characterize the efficiency of our proposed model, we perform a simple experimental check. In Tab. 4, we compare the total training time needed for different Graph-SSL strategies to provide a representation that performs optimally on the downstream task. We show results on the datasets with the largest graphs from Tab. 1: IMDB and REDDIT-M5. Since runtime is hardware-dependent, all experiments are performed on the same machine. Graph-JEPA displays superior efficiency and promising scaling behavior. The presented runtime is naturally dependent on the self-supervised scheme used in each model, so we do not regard it as a definitive descriptor but rather an indicator of the potential of entirely latent self-predictive models.

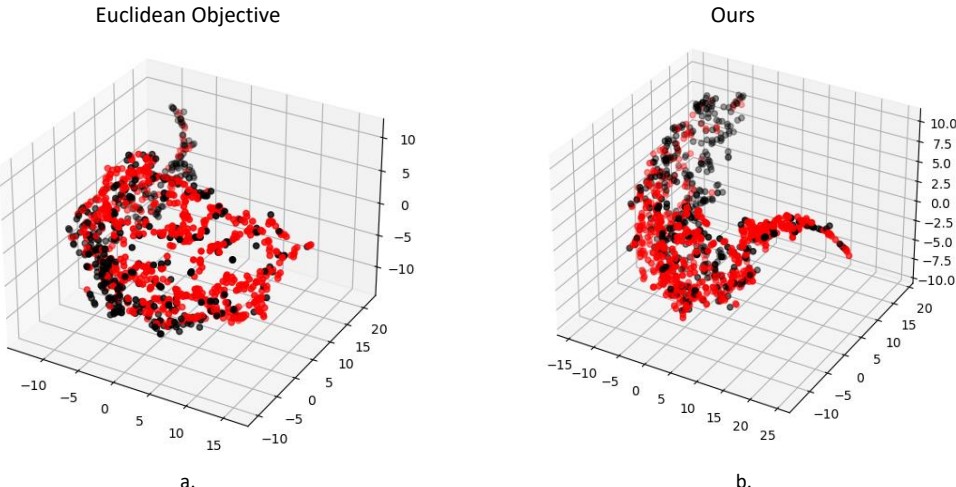

Figure 3: 3D t-SNE(Van der Maaten & Hinton, 2008) of the latent representations used to train the linear classifier on the DD dataset. The change in the curvature of the embedding using the Graph-JEPA objective (b.) is noticeable. Best viewed in color.

Table 3: Comparison of Graph-JEPA performance for different distance functions. The optimization for Poincaré embeddings in higher dimensions is problematic, as shown by the *NaN* loss on the IMDB-B dataset. LD stands for Lower Dimension, where we use a smaller embedding size. The **best result** is reported in **boldface**.

| Dataset | Ours | Euclidean | Hyperbolic | Euclidean (LD) | Hyperbolic (LD) |
|---|---|---|---|---|---|
| MUTAG | **91.25 ± 5.75** | 87.04 ± 6.01 | 89.43 +- 5.67 | 86.63 ± 5.9 | 86.32 ± 5.52 |
| REDDIT-M | **56.73 ± 1.96** | 56.55 ± 1.94 | 56.19 +- 1.95 | 54.84 ± 1.6 | 55.07 ± 1.83 |
| IMDB-B | 73.68 ± 3.24 | **73.76 ± 3.46** | *NaN* | 72.5 ± 3.97 | 73.4 ± 4.07 |
| ZINC | **0.434 ± 0.01** | 0.471 ± 0.01 | 0.605 +- 0.01 | 0.952 ± 0.05 | 0.912 ± 0.04 |

**MLP Parametrization.** Tab. 5 contains the results of parametrizing the whole architecture, other than the initial GNN encoder, through Multilayer Perceptrons (MLPs). This translates to not using the Attention mechanism at all. For this experiment, and also the following ablations, we consider 4 out of the 8 datasets initially presented in Tab. 1, making sure to choose different graph domains for a fair comparison. Graph-

Table 4: Total training time and model parameters of MVGRL, GraphMAE, and Graph-JEPA for pretraining (single run) based on the optimal configuration for downstream performance. OOM stands for Out-Of-Memory. The **best result** is reported in **boldface**.

| Dataset | Model | Num. parameters | Training time |
|---|---|---|---|
| IMDB | MVGRL | 3674118 | ∼ 7 min |
| | GraphMAE | 2257193 | ∼ 1.5 min (1min 36sec) |
| | Graph-JEPA | **19219460** | **< 1min (56 sec)** |
| REDDIT-M5 | MVGRL | 4198406 | *OOM* |
| | GraphMAE | 2784555 | ∼ 46 min |
| | Graph-JEPA | **19245060** | **∼ 18 min** |

Table 5: Performance when parametrizing the context and target encoders through MLPs vs using the proposed Transformer encoders. The **best result** is reported in **boldface**.

| Dataset | Transformer Encoders | MLP Encoders |
|---|---|---|
| MUTAG | **91.25 ± 5.75** | 90.5 ± 5.99 |
| REDDIT-M5 | **56.73 ± 1.96** | 56.21 ± 2.29 |
| IMDB-B | 73.68 ± 3.24 | **74.26 ± 3.56** |
| ZINC | **0.434 ± 0.01** | 0.472 ± 0.01 |

Table 6: (a) Performance when extracting subgraphs with METIS vs. using random subgraphs. (b) Performance when using node-level vs patch-level RWSEs. The **best result** is reported in **boldface**.

(a)

| Dataset | METIS | Random |
|---|---|---|
| MUTAG | 91.25 ± 5.75 | **91.58 ± 5.82** |
| REDDIT-M5 | **56.73 ± 1.96** | 56.08 ± 1.69 |
| IMDB-B | **73.68 ± 3.24** | 73.52 ± 3.08 |
| ZINC | 0.434 ± 0.01 | **0.43 ± 0.01** |

(b)

| Dataset | Node RWSE | Patch RWSE |
|---|---|---|
| MUTAG | **91.25 ± 5.75** | 91.23 ± 5.86 |
| REDDIT-M5 | **56.73 ± 1.96** | 56.01 ± 2.1 |
| IMDB-B | **73.68 ± 3.24** | 73.58 ± 4.47 |
| ZINC | **0.434 ± 0.01** | 0.505 ± 0.005 |

JEPA still manages to perform well, showing the flexibility of our architecture, even though using the complete Transformer encoders leads to better overall performance and less variance in the predictions.

**Positional Embedding.** Following He et al. (2023), it is possible to use the RWSE of the patches as conditioning information. Formally, let $B \in \{0,1\}^{p \times N}$ be the patch incidence matrix, such that $B_{ij} = 1$ if $v_j \in p_i$. We can calculate a coarse patch adjacency matrix $A' = BB^T \in \mathbb{R}^{p \times p}$, where each $A'_{ij}$ contains the node overlap between $p_i$ and $p_j$. The positional embedding can be calculated for each patch using the RWSE described in Eq. 1 on $A'$. We call such an embedding relative, as it can only capture positional differences in a relative manner between patches. We test Graph-JEPA with these relative positional embeddings and find that they still provide good performance but consistently fall behind the node-level (global) RWSE that we employ in our formulation (Tab. 6b). An issue of these relative patch RWSEs is that the number of shared neighbors can obscure the local peculiarities of each patch, rendering the context given to the predictor more ambiguous.

**Random Partitioning.** A natural question in our framework is how to design the spatial partitioning procedure. Using a structured approach like METIS (Karypis & Kumar, 1998) is intuitive and leads to favorable results. Another option would be to extract random, non-empty subgraphs as context and targets. As seen in Tab. 6a, the random patches also provide strong performance, showing that the proposed JEPA architecture is not reliant on the initial input, which is the case for many methods that rely on data augmentation for view generation (Lee et al., 2022). Even though our results show that using a structured way to extract the patches might not be necessary, it is an idea that generalizes well to different graph types and sizes. Thus, we advocate extracting subgraphs with METIS as it is a safer option in terms of generalizability across different graphs and the inductive bias it provides.

## 5  Conclusion

In this work, we introduce a new Joint Embedding Predictive Architecture (JEPA) for graph-level Self-Supervised Learning (SSL). An appropriate design of the model, both in terms of data preparation and pretraining objective, reveals that it is possible for a neural network to self-organize the semantic knowledge embedded in a graph, demonstrating competitive performance in different graph data and tasks. Future research directions include extending the proposed method to node and edge-level learning, theoretically exploring the expressiveness of Graph-JEPA, and gaining more insights into the optimal geometry of the latent space for general graph SSL.

**Acknowledgments**

Geri Skenderi is funded by the European Union through the Next Generation EU - MIUR PRIN PNRR 2022 Grant P20229PBZR. The views and opinions expressed are, however, those of the authors only and do not necessarily reflect those of the European Union or the European Research Council Executive Agency. Neither the European Union nor the granting authority can be held responsible.

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

# 6 Appendix

## 6.1 Dataset Statistics and Model Hyperparameters

Table 7: Descriptive statistics of the TUD datasets used for the main experiments.

| Dataset | Num. Graphs | Num. Classes | Avg. Nodes | Avg. Edges |
|---------|-------------|--------------|------------|------------|
| PROTEINS | 1113 | 2 | 39.06 | 72.82 |
| MUTAG | 188 | 2 | 17.93 | 19.79 |
| DD | 1178 | 2 | 284.32 | 715.66 |
| REDDIT-B | 2000 | 2 | 429.63 | 497.75 |
| REDDIT-M5 | 4999 | 5 | 508.52 | 594.87 |
| IMDB-B | 1000 | 2 | 19.77 | 96.53 |
| IMDB-M | 1500 | 3 | 13.00 | 65.94 |
| ZINC | 12000 | 0 | 23.2 | 49.8 |

Table 8: Values of Graph-JEPA specific hyperparameters for each dataset.

| Hyperparameter | PROTEINS | MUTAG | DD | REDDIT-B | REDDIT-M5 | IMDB-B | IMDB-M | ZINC |
|----------------|----------|-------|-----|----------|-----------|--------|--------|------|
| Num. Subgraphs | 32 | 32 | 32 | 128 | 128 | 32 | 32 | 32 |
| Num. GNN Layers | 2 | 2 | 3 | 2 | 2 | 2 | 2 | 2 |
| Num. Encoder Blocks | 4 | 4 | 4 | 4 | 4 | 4 | 4 | 4 |
| Embedding size | 512 | 512 | 512 | 512 | 512 | 512 | 512 | 512 |
| RWSE size | 20 | 15 | 30 | 40 | 40 | 15 | 15 | 20 |
| Num. context - Num. targets | 1 - 2 | 1 - 3 | 1 - 4 | 1 - 4 | 1 - 4 | 1- 4 | 1- 4 | 1- 4 |

## 6.2 Additional Figures

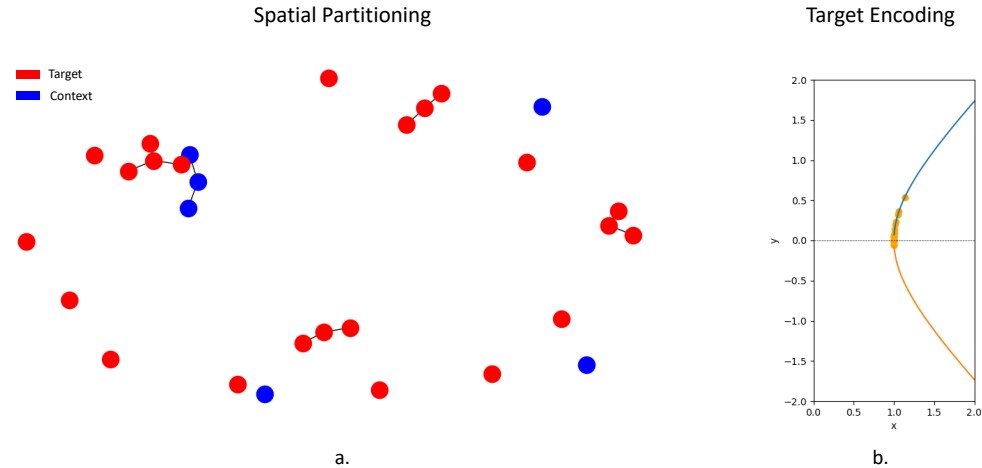

Figure 4: Visualization of the partition of a small graph from the MUTAG dataset used as input (a.) and the corresponding (learned) target embeddings in the hyperbolic plane by Graph-JEPA (b.) Best viewed in color.

### 6.3 Additional Experimental Results

To gain additional empirical insight into Graph-JEPA and its components, we evaluate the model in two complex long-range tasks from the LRGB benchmark (Dwivedi et al., 2022), consisting of predictive tasks over short chains of amino acids called peptides:

- Peptides-struct is a multi-label graph regression dataset based on the 3D structure of the peptides. It can, therefore, be treated as a multivariate regression problem where the dependent variable is 11-dimensional.

- Peptides-func is a multi-label graph classification dataset, given that a peptide can be considered to belong to several classes simultaneously. There are 10 labels in total, and they are imbalanced. It consists of the same graphs as Peptides-struct, therefore it gives us the ability to evaluate how good the representation learning model is in learning specific downstream tasks.

For both datasets, we follow the work of Dwivedi et al. (2022) and perform four runs with different seeds. The results are presented in Tab. 9. To have a baseline against which to compare, we also list both GNN and Transformer-based models trained in a supervised fashion. Two main takeaways arise:

1. The design choices of the positional embedding (Sec. 3.2) and the proposed self-predictive task (Sec. 3.4) are both beneficial for downstream performance, even in long-range tasks.

2. The type of downstream task can significantly affect the judgment of how good the learned representations are. Remember that both datasets in Tab. 9 present the same graphs, yet the regression task shows highly competitive performance. In contrast, the multi-label classification task presents an enormous drop in performance. We argue that this is due to the nature of the task. Considering how little correlation there is between the labels, localized, specific interactions are key to achieving optimal performance. This was also reported by Dwivedi et al. (2022), where the authors showed how using a Transformer with the fully connected graph as input would provide optimal performance. Graph-JEPA excels at capturing smooth global features (helpful for Peptides-struct) but fails to capture localized, specific interactions (crucial for Peptides-func) due to the nature of the subgraph self-predictive task. Nevertheless, the previous observation regarding the value of the different elements of our design holds even in the case when downstream performance is suboptimal.

We hope these additional results can inspire future research directions in JEPAs and graph SSL, as we are still far away from an optimal and general architecture that can learn well across different graph types and tasks.

Table 9: (a) Performance of Graph-JEPA compared to supervised baselines on the two peptide datasets from the Long Range Graph Benchmark (LRGB). (a) Graph regression performance (3D properties of peptides) (b) Multilabel graph classification performance (peptide function). The **best results** are reported in **boldface**, and the second best are underlined. The baselines are taken from Dwivedi et al. (2022), and they *are all supervised models*. † represents our model without the PE described in Sec. 3.2, while ‡ is both without the PE and using the Euclidean latent objective. The results are reported over four runs with different seeds, as commonly done in the literature.

(a)

| Model | Test MAE ↓ | Train R2 ↑ |
|---|---|---|
| GCN | $0.350 \pm 0.001$ | $0.651 \pm 0.008$ |
| GatedGCN + RWSE | $0.3357 \pm 0.001$ | $0.720 \pm 0.015$ |
| SAN + RWSE | $\mathbf{0.2545 \pm 0.001}$ | $0.711 \pm 0.005$ |
| Graph-JEPA ‡ | $0.313 \pm 0.003$ | $0.703 \pm 0.002$ |
| Graph-JEPA † | $0.308 \pm 0.002$ | $\underline{0.73 \pm 0.005}$ |
| Graph-JEPA | $\underline{0.305 \pm 0.002}$ | $\mathbf{0.731 \pm 0.003}$ |

(b)

| Model | Test AP ↑ |
|---|---|
| GCN | $0.593 \pm 0.002$ |
| GatedGCN + RWSE | $\underline{0.607 \pm 0.004}$ |
| SAN + RWSE | $\mathbf{0.644 \pm 0.008}$ |
| Graph-JEPA † | $0.24 \pm 0.002$ |
| Graph-JEPA ‡ | $0.252 \pm 0.002$ |
| Graph-JEPA | $0.263 \pm 0.003$ |

