# OpenReview forum: "Graph-level Representation Learning with Joint-Embedding Predictive Architectures"
_TMLR — Accepted by TMLR_

### Review · Reviewer_pezm · 2024-08-25

**Summary Of Contributions:**

This paper extends the idea of  Joint-Embedding Predictive Architectures (JEPAs), often seen in Images and other domains, to graphs. The key idea is to predict representations of the masked subgraphs from that of context subgraphs. Experiments show some improvement over the existing methods.

**Audience:**

Yes

**Claims And Evidence:**

Yes

**Requested Changes:**

1. First of all, the authors should list their contributions clearly, which positions them uniquely in the field.

2. The proposed method doesn't have anything specific to graphs. It would be nice if the Authors could highlight if any components are unique to graphs.

3. Figure 2 should be improved to convey more information. It looks, toy-like now, with very little information.

4. The assumptions made in Eq. 7 are over-simplistic (linearity of predictor networks). Authors should clarify more on what happens if these are relaxed.

5. In Table 3, the performance of all the previous methods is within the error bars of the proposed method. How could this be explained?

6. Did the Authors consider ablating their method by removing different components?

7. I didn't understand what the tsne plot in Fig. 3 conveys. In addition, why is the classifier considered linear there? You could consider plotting tsne for the penultimate layer of a deeper NN as well.

**Strengths And Weaknesses:**

Strengths:

1. This seems to be the first work to apply JEPA on graphs.

2. Experiments seem to suggest some improvement.

Weaknesses:

1. The biggest weakness of this method is that it seems like a straightforward extension of JEPA on graphs.

2. The above weakness can be overlooked if the method offers significant advantages  over the existing methods. However, the improvement seems marginal, given that, the question of significance comes up.

3. Lack of any deep theoretical insights/analysis.

---

> ### Author Response · Authors · 2024-08-30
> **Initial response to Reviewer pezm**
>
> We would like to thank the reviewer for the constructive suggestions and for acknowledging the potential of our work as the first to apply the JEPA paradigm to graphs. We appreciate the feedback and provide a response to each of the requested changes below, with the aim of further clarifying and enhancing our contributions.
>
> > 1. First of all, the authors should list their contributions clearly, which positions them uniquely in the field.
>
> We recognize that there is no list summarizing our contributions in the current manuscript, but we opted for this choice as a matter of writing. The final two paragraphs in the introduction discuss our contributions and their uniqueness in depth. We believe these can be briefly summarized in the following three points The list below offers a summary of these contributions:
>
> - We proposed the first JEPA, based on the recent formalization of LeCun [1], in the graph domain for graph-level representation learning.
>
> - In order to learn representations that reflect the hierarchical nature of concepts encoded in graph-level labels, we designed an approximate hyperbolic self-predictive training objective, which is simple to calculate and is shown empirically to perform well. Furthermore, to the best of our knowledge, this objective is the only one that can readily be transferred to other works that wish to perform self-predictive graph-level self-supervised learning.
>
> - The proposed JEPA is extremely flexible, as the various ablations show. Furthermore, it is also computationally efficient when compared to other graph SSL paradigms, opening the way for further research in this new self-supervised learning paradigm on graphs.
>
>
>
>
> > 2. The proposed method doesn't have anything specific to graphs. It would be nice if the Authors could highlight if any components are unique to graphs.
>
> There are several components of our design that are specific to graphs. Starting from the more minor yet important details, the view generation and their encoding are graph-specific: We generate subgraphs based on a graph partitioning procedure and then use a GNN to learn features for each subgraph. What can be considered the key contribution of the proposed model is the (approximate) hyperbolic self-predictive objective, which was implemented with graph-level tasks in mind (where hierarchy is often present [2,3,4]) and is thus very specific to this data domain. We consider the flexibility of the encoder and predictor networks to be a big advantage of Graph-JEPA. However, we still show that the best performance is achieved when using a graph-specific attention module rather than a straightforward and more general MLP parametrization. Based on all the above, we believe that in all components that make up Graph-JEPA, there graph-specific aspects. It is worth noting that all of these are inherited from the more general principles of the JEPA SSL paradigm [1], which are finding widespread use in a vast array of data modalities [5,6,7].
>
> > 3. Figure 2 should be improved to convey more information. It looks, toy-like now, with very little information.
>
> We would like to thank the reviewer for this consideration. Improving the informativeness of Figure 2 will be a significant focus for the revised version of the manuscript. Nevertheless, at this point of the review process, as recommended by the TMLR guidelines, we will not make any changes to the Figure in the main manuscript while waiting for the other reviews to facilitate the reviewing process. The initial idea for the revised figure is to consider a smaller input graph, such that only one context and target patch are present, therefore leaving the possibility to visually explain every single component of the architecture more detailed.
>
> > 4. The assumptions made in Eq. 7 are over-simplistic (linearity of predictor networks). Authors should clarify more on what happens if these are relaxed.
>
> We agree with the reviewer that having the last part (i.e., parts e. and f. in Figure 2) of the proposed architecture be linear is an overly simplistic modeling assumption. Nevertheless, as we mention in the theoretical analysis that follows, even such a simple scenario would lead to representation collapse, which justifies the use of a training trick such as the stop gradient operation and EMA weight update (also often seen in other works in Self-Supervised Learning [BYOL]). Therefore, choosing a simple scenario and showing that problems would arise, even in that case, is a deliberate choice. While a more detailed theoretical analysis in the non-linear case would be pretty valuable, it is both beyond the scope of this work and also very challenging, considering the non-local propagation of information and the nonlinearity of all the neural networks present in the architecture.

---

> > ### Author Response · Authors · 2024-08-30
> > **Continuation of initial response to Reviewer pezm**
> >
> > > 5. In Table 3, the performance of all the previous methods is within the error bars of the proposed method. How could this be explained?
> >
> > We conjecture that there are three main reasons for the variance in the results:
> >
> > - The view generation consists of a subgraph partitioning procedure, which in turn is quite dependent on randomness. Therefore, if trivial subgraphs are chosen as context subgraphs, it becomes very difficult to predict the latent representations of the target subgraphs.
> >
> > - The advantages of the JEPA paradigm are numerous, even in the graph domain, as we have shown in our manuscript. Nevertheless, by predicting the latent representations and working in latent space only, the models are very susceptible to representation collapse, as we have also discussed in the manuscript. In cases where the positional embeddings of the target subgraphs are very similar, for example, smaller graphs, it is likely that the model produces very similar representations for all subgraphs, thus rendering the prediction task easy but degrading downstream effectiveness. We have not performed extensive hyperparameter tuning, as the main focus was to show that our proposed architecture can perform well using a relatively simple training recipe. This is also reflected in the hyperparameter values in Table 1.
> >
> > - Last but not least, when implementing the evaluation procedure in practice, many self-supervised learning models follow a protocol to train the model on the full dataset (testing set included) and then simply finetune the linear classifier on the training set. We find this evaluation protocol to be somewhat unfair and not a great indicator of generalization, and train the self-supervised model only on the training set to be as fair as possible. This is also the case of BGRL, where the same phenomenon can be seen. While this last part is likely not the root cause, we believe it does contribute, but it is also the fairest option.
> >
> > The points above are important takeaways, and we plan to include a summary of them in the experimental section, as they are currently missing from the manuscript. Therefore, we would like to thank the reviewer for raising this point.
> >
> > > 6. Did the Authors consider ablating their method by removing different components?
> >
> > The experimental section contains different such ablations. One clear example is removing the attention mechanism in the encoder networks and using MLPs, while another is not using a graph clustering algorithm for the partitioning but performing random partitioning (i.e., removing the initial view generation logic). Finally, we also provide thorough experiments by removing the proposed self-predictive objective and replacing it with the simple and widely spread Mean Squared Error objective. We consider these to be important ablations that demonstrate the important capabilities of Graph-JEPA. The rest of the architectural details are either more difficult or trivial to ablate, which is why we have not added any other empirical studies to the paper.

---

> > > ### Author Response · Authors · 2024-08-30
> > > **Continuation (2) of initial response to Reviewer pezm**
> > >
> > > > 7. I didn't understand what the tsne plot in Fig. 3 conveys. In addition, why is the classifier considered linear there? You could consider plotting tsne for the penultimate layer of a deeper NN as well.
> > >
> > > The purpose of Figure 3 is to show that the geometry of the latent representations is strongly affected by the self-supervised learning objective. The plots provided are indeed the same as when plotting the representations of the penultimate layer of a deep NN (for example, a deep CNN), with the last layer being the classifier. When using the mean squared error objective in high dimensions, the more spherical and uniform data manifold is shown in Figure 3a. Is created. While using our proposed objective, the data manifold becomes more hyperbolic, as can be qualitatively seen in Figure 3b. This directly impacts classification performance. The choice of the linear classifier is standard practice in the Self-Supervised Learning literature throughout all data domains ([8,9]), a procedure often referred to as Linear Probing. This allows us to see how semantic and representative the features learned by the network are. Using complicated and non-linear downstream classifiers defeats the purpose of evaluating the goodness of the learned features, as the classifier can then learn useful features by itself.
> > >
> > > [1] Dawid, Anna, and Yann LeCun. "Introduction to latent variable energy-based models: A path towards autonomous machine intelligence." arXiv preprint arXiv:2306.02572 (2023).
> > >
> > > [2] Li, Jia, et al. "Semi-supervised hierarchical graph classification." IEEE Transactions on Pattern Analysis and Machine Intelligence 45.5 (2022): 6265-6276.
> > >
> > > [3] Chami, Ines, et al. "Hyperbolic graph convolutional neural networks." Advances in neural information processing systems 32 (2019).
> > >
> > > [4] Ying, Zhitao, et al. "Hierarchical graph representation learning with differentiable pooling." Advances in neural information processing systems 31 (2018).
> > >
> > > [5] Fei, Zhengcong, Mingyuan Fan, and Junshi Huang. "A-jepa: Joint-embedding predictive architecture can listen." arXiv preprint arXiv:2311.15830 (2023).
> > >
> > > [6] Assran, Mahmoud, et al. "Self-supervised learning from images with a joint-embedding predictive architecture." Proceedings of the IEEE/CVF Conference on Computer Vision and Pattern Recognition. 2023.
> > >
> > > [7] Bardes, Adrien, Jean Ponce, and Yann LeCun. "Mc-jepa: A joint-embedding predictive architecture for self-supervised learning of motion and content features." arXiv preprint arXiv:2307.12698 (2023).
> > >
> > > [8] Alain, Guillaume, and Yoshua Bengio. "Understanding intermediate layers using linear classifier probes." arXiv preprint arXiv:1610.01644 (2016).
> > >
> > > [9] Grill, Jean-Bastien, et al. "Bootstrap your own latent-a new approach to self-supervised learning." Advances in neural information processing systems 33 (2020): 21271-21284.

---

### Review · Reviewer_XWJw · 2024-09-03

**Summary Of Contributions:**

This paper introduces a novel approach for graph-level representation learning using Joint-Embedding Predictive Architectures (JEPAs). The authors propose Graph-JEPA, which leverages masked modeling to predict the latent representations of masked subgraphs from context subgraphs. The experimental results highlight the effectiveness of Graph-JEPA in various tasks.

**Audience:**

Yes

**Claims And Evidence:**

Yes

**Requested Changes:**

To provide a clearer picture of the experiment scale, it would be beneficial to explicitly state the size of the datasets used, including details such as graph dimensions, the number of samples, etc.

More detailed explanations and interpretations of the numerical results are needed. For example, in Table 2 (Proteins), while Graph-JEPA shows the second-best result in terms of the mean, its standard deviation is significantly higher -- ten times that of S2GAE (the best for this dataset) and GraphMAE, LaGraph, which exhibit similar performances in terms of average values. This raises questions about whether Graph-JEPA can truly be considered the second-best in this context, and it would be helpful to provide further elaboration on its higher variance.

Minor: There is a typo in the caption of Table 7; (a) and (b) should be swapped.

**Strengths And Weaknesses:**

Strengths:
1. The proposed Graph-JEPA introduces a novel way to apply JEPAs to graph data, providing an alternative to contrastive and generative models that avoids the need for negative (or positive) samples and complex data augmentation.
2. The paper provides extensive experimental results across multiple datasets, showing that Graph-JEPA outperforms existing methods in terms of both performance and efficiency. The detailed ablation studies further support the robustness of the proposed approach.

Weakness:
Clarity of Presentation: Some sections of the paper could benefit from clearer explanations, particularly around the rationale behind choosing specific architectural components, see below.

---

> ### Author Response · Authors · 2024-09-04
> **Initial response to Reviewer XWJw**
>
> We would like to thank the reviewer for acknowledging the novelty and efficacy of our proposed approach. An initial response to the requested changes is provided below.
>
> > To provide a clearer picture of the experiment scale, it would be beneficial to explicitly state the size of the datasets used, including details such as graph dimensions, the number of samples, etc.
>
> We opted not to include this information initially in the manuscript simply for space reasons, but the reviewer is correct in that without the details of the datasets, it becomes hard to understand the scale. Therefore, an additional table containing details on the TUD datasets will be added in the future revised manuscript. At this point of the review process, as recommended by the TMLR guidelines, we will not make any changes to the main manuscript while waiting for the other reviews to facilitate the reviewing process.
>
> > More detailed explanations and interpretations of the numerical results are needed. For example, in Table 2 (Proteins), while Graph-JEPA shows the second-best result in terms of the mean, its standard deviation is significantly higher -- ten times that of S2GAE (the best for this dataset) and GraphMAE, LaGraph, which exhibit similar performances in terms of average values. This raises questions about whether Graph-JEPA can truly be considered the second-best in this context, and it would be helpful to provide further elaboration on its higher variance.
>
> We conjecture that there are three primary reasons for the variance in our results:
> 1. The view generation process involves subgraph partitioning, which is heavily influenced by randomness. Consequently, if trivial subgraphs are selected as context subgraphs, it becomes challenging to accurately predict the latent representations of the target subgraphs.
>
> 2. The JEPA paradigm offers numerous benefits, even in the graph domain, as demonstrated in our manuscript. However, since the models operate solely in latent space by predicting latent representations, they are prone to representation collapse, as discussed in our manuscript. This issue is particularly pronounced when the positional embeddings of the target subgraphs are very similar, such as with smaller graphs. In such cases, the model tends to generate nearly identical representations for all subgraphs, making the prediction task easier but reducing downstream effectiveness. Our primary goal was to demonstrate the efficacy of our proposed architecture with a straightforward training regimen. Hence, we did not conduct extensive hyperparameter tuning. This is reflected in the hyperparameter values presented in Table 1.
>
> 3. In practical implementations of the evaluation procedure, many self-supervised learning models adhere to a protocol where the model is trained on the entire dataset (including the testing set) before finetuning the linear classifier on the training set. We believe this protocol is somewhat unfair and does not accurately indicate generalization. Therefore, we train the self-supervised model exclusively on the training set to ensure fairness. This approach is also used for BGRL, where a similar phenomenon is observed. Although this factor may not be the primary cause of variance, it contributes to the overall fairness of the evaluation.
> Despite the high variance, the average case performance is the most important aspect of the experimental validation. That is because, by considering the variance, one then has to consider best and worst case scenarios, where, for example, the best case scenario of Graph-JEPA on Proteins would outperform the other best cases by a large margin. We will make sure to include a synthesized version of the above in the revised version of the manuscript, given that, as the reviewer pointed out, some details are missing.
>
> > Minor: There is a typo in the caption of Table 7; (a) and (b) should be swapped.
>
> We would like to thank the reviewer for the keen observation, there is indeed a typo in the caption of Tab. 7. We will fix this in the revision.

---

### Review · Reviewer_dx84 · 2024-10-10

**Summary Of Contributions:**

This work investigates extending recent advances in self-supervised learning, in particular the JEPA model, to the scenario of graph data. The methodology is mostly aligned with the image SSL approach presented by I-JEPA. In order to apply JEPA to graphs, several adaptations are needed. The first is to define a subgraph partitioning strategy so that a full graph can be broken down into (mostly) disjoint subgraphs, similar to how images are processed in separate patches by ViT architectures. The next is to define a RWSE positional encoding strategy for subgraphs to provide the model with context about how the subgraphs are connected, similar to positional encoding for image patches. Finally, the L2 loss function for Graph JEPA is performed in a hyperbolic space which is believed to better preserve hierarchical graph information. Experiments show that the proposed method can achieve strong performance in downstream graph classification and regression tasks, performing on par with or better than a variety of related methods across standard benchmark datasets.

**Audience:**

Yes

**Broader Impact Concerns:**

Ethical implications were not discussed, although this does not impact my assessment of this work.

**Claims And Evidence:**

Yes

**Requested Changes:**

I request that the authors respond to my questions in the weakness section to provide further background about the method.

**Strengths And Weaknesses:**

Strengths:
* The motivation and presentation of this work is very clear and intuitive.
* Extending JEPA methodology to graph data is a promising and interesting direction. The methods used to adapt JEPA for the graph space (subgraph splitting analogous to image patching, positional embedding with RWSE, loss in hyperbolic space) are intuitive and natural.
* Experiments show strong performance of the proposed method compared to related graph SSL methods.
* Design choices are thoroughly ablated, including the use of hyperbolic loss, GNN parameterization, local vs. global positional embedding, and method for subgraph selection/patching.

Weaknesses
* One primary weakness of the proposed method is the potential loss of important connection information when the graph is broken into subgraphs. It seems that connections at any level higher than the subgraph connections will not be able to influence the model directly (the only mechanism for this seems to be the positional embedding). In cases where the connectivity between subgraphs is high and specific subgraph connections provide important information, the modeling capacity of the proposed method could be limited.
* A major difference between the proposed method and I-JEPA is that I-JEPA will use multiple patches as context, while the proposed method only uses a single subgraph (similar to a single image patch) as context. Is there a reason for this choice? Would including multiple subgraphs as context enhance the performance of the model? Using a relatively large number of context patches seems to be an important aspect of I-JEPA.
* The paper claims "we would ideally like to have a high-dimensional latent code that has a concept of hyperbolicity built into it", but in the end the latent code for each target subgraph is only 2D. It would be helpful to clarify this.
* (minor) What is the matrix D on page 5 under eq. (1)?

---

> ### Author Response · Authors · 2024-10-16
> **Initial response to Reviewer dx84**
>
> We sincerely appreciate the reviewers' recognition of the presentation of our work and its potential contribution in adapting JEPA to graphs. Below, we address the listed concerns, with the aim of providing as much clarity as possible:
>
> > One primary weakness of the proposed method is the potential loss of important connection information when the graph is broken into subgraphs. It seems that connections at any level higher than the subgraph connections will not be able to influence the model directly (the only mechanism for this seems to be the positional embedding). In cases where the connectivity between subgraphs is high and specific subgraph connections provide important information, the modeling capacity of the proposed method could be limited.
>
> It is natural to consider such a plausibly problematic scenario, given the patch-based procedure of our model. To address this issue in our design, we take three deliberate steps:
>
> 1. Subgraph Creation: We explicitly advocate for the use of METIS [1], a graph clustering algorithm that partitions the graph based on the principle that within-cluster links are much denser than between-cluster links. This allows each subgraph to retain 'community information,' which strengthens the subgraph-level connections, as the reviewer mentioned. This can be an important factor in certain, types of graphs.
> 2. Subgraph Expansion: To introduce a more global context, we expand the subgraphs using a 1-hop neighbor expansion. This creates overlap between subgraphs, which, when combined with the message propagation in the GNN encoder, ensures that neighboring subgraphs share common features.
> 3. Positional Encoding: This step provides the most global information, which is then used as conditioning to aid in the self-predictive task.
>
> These steps enable us to rely on the proposed self-predictive pretext task, which remains challenging but not unfeasible for learning. Due to steps 2 and 3, subgraphs with strong connectivity are easier to predict, while subgraphs that are farther apart (in terms of shortest path distance) become more difficult to predict. This is an intended design choice for a JEPA architecture, as overly simple self-predictive tasks can cause models to suffer from representation collapse early in training.
>
> > A major difference between the proposed method and I-JEPA[2] is that I-JEPA will use multiple patches as context, while the proposed method only uses a single subgraph (similar to a single image patch) as context. Is there a reason for this choice? Would including multiple subgraphs as context enhance the performance of the model? Using a relatively large number of context patches seems to be an important aspect of I-JEPA.
>
> The reviewer is correct in highlighting the importance of the number of context patches in I-JEPA [2]. Our response here closely relates to the previous one, as a key distinction is that in I-JEPA, the authors explicitly state that 'there is no overlap between the context and target blocks in all considered strategies' (page 8, below Table 7). This is due to the nature of image data, where locality is a useful prior. In our case, two main factors are what justify the design choice of the single context block:
> - The neighborhood expansion at the subgraph level creates an overlap, which when combined with the positional embedding, enables effective self-prediction.
> - Partitioning graph data into overlapping subgraphs means using multiple context patches could result in overly simplistic scenarios.
> Nonetheless, this is an exciting direction for future work, particularly on very sparse graphs. Introducing more context patches could be valuable if a method were found to regularize the feature overlap between patches, ensuring they don’t cover the entire graph and make the self-predictive task trivial.
>
> > The paper claims "we would ideally like to have a high-dimensional latent code that has a concept of hyperbolicity built into it", but in the end the latent code for each target subgraph is only 2D. It would be helpful to clarify this.
>
> In our methodology, the latent code refers to the embedding vectors $z^x$, which is used to generate the hyperbolic angle. The embedding size we use in practice is (relatively) large but does not encounter the issues typically associated with high-dimensional hyperbolic latent embeddings, as discussed in the paper. Simply put, we work with a high-dimensional latent code that is transformed into a 2D coordinate solely for the self-predictive task. However, for downstream tasks, we continue using the high-dimensional latent code, which inherently captures the concept of hyperbolicity, as it is the basis for generating the hyperbolic coordinates.

---

> > ### Author Response · Authors · 2024-10-16
> > **Continuation of initial response to Reviewer dx84**
> >
> > > (minor) What is the matrix D on page 5 under eq. (1)?
> >
> > We want to thank the reviewer for this keen observation. The matrix $D$ is an $N \times N$ diagonal matrix containing each node's degree. By using the equation where $D$ appears, we can row-normalize the adjacency matrix $(D^{-1}A)$, so that each row sums to $1$, effectively representing a distribution over its neighboring nodes. This is why M can be interpreted as the random walk transition matrix. We will add this detail in the revised version of the paper.
> >
> > [1] George Karypis and Vipin Kumar. "A fast and high quality multilevel scheme for partitioning irregular graphs." SIAM Journal on scientific Computing, 20(1):359–392, 1998
> >
> > [2] Assran, Mahmoud, et al. "Self-supervised learning from images with a joint-embedding predictive architecture." Proceedings of the IEEE/CVF Conference on Computer Vision and Pattern Recognition. 2023.

---

### Author Response · Authors · 2024-10-16
**Paper revision**

Dear Reviewers,

Thank you for the valuable feedback and for the timely review process. With this comment, we'd like to notify you that we have revised some aspects of the paper based on your comments. The changes in the revised manuscript compared to the submitted version are *highlighted in blue*. Specifically, we have changed the following:

- Fig. 1 has been modified to look less crowded and contain more details in the caption, as entailed by the feedback from reviewer pezm. Along with the figure, we fixed any parts of the text referencing it so that they would fit with the new one.
- Added a new table (Table 1) containing information about the datasets used for the experiments, as suggested by reviewer XWJw.
- Added a paragraph explaining possible reasons for the variance in the results based on the feedback and response from reviewer XWJw.
- Fixed the typo in Table 7 (now Table 8), as suggested by reviewer XWJw.
- Added a brief mention of D being the diagonal degree matrix when talking about the embedding method, as suggested by reviewer dx84

---

### Author Response · Authors · 2025-01-18
**Final manuscript**

Dear AE, Reviewers, and EiCs,

We are uploading, alongside this comment, the revised manuscript to be sent for publication. Firstly, we greatly thank the reviewers and the AE for the enjoyable and informative review process. We truly believe it helped elevate the quality of the manuscript. In this final version, we have made several changes, listed in the following:

- An appendix was added. The tables containing the dataset statistics and model hyperparameters, which were previously situated in the main paper, have been placed in the appendix. In said appendix, we provide new experimental results consisting of Graph-JEPA ablations on long-range tasks from the Long Range Graph Benchmark [1], both in multivariate graph regression and multilabel classification. Finally, we provide a figure showing the context and target patches from a small molecular graph in Fig. 4 in the Appendix.

- We used the free space remaining in the main manuscript to expand significantly the details on the spatial partitioning and subgraph positional embedding, as requested in the first three bullet items requested by the AE in point 1. of their decision.

- Minor details on the latent codes and the stochasticity regarding the subgraph partitioning were added.

- Uniformized the notation and typography of the paper, trying to pay close attention to the language used and the grammatical structure.

- Last but not least, we added the official code repository (Github) and deanonymized the paper to be ready for publication

[1] Dwivedi, Vijay Prakash, et al. "Long range graph benchmark." Advances in Neural Information Processing Systems 35 (2022): 22326-22340.

---

### Decision · Action_Editor_jQqc · 2024-12-08

**Recommendation:** Accept with minor revision

**Comment:**

After a thorough evaluation of the manuscript and the authors’ responses to the reviewers’ critiques, I recommend that the paper be accepted for publication, contingent upon the careful implementation of the suggested by reviewers improvements. In particular:
1) Elaborate on global context mechanisms:
-   Provide a more detailed discussion of how community detection-based partitioning (e.g., METIS) ensures that each subgraph retains meaningful structural information. Include a formal definition and a brief theoretical justification as to why methods like METIS lead to subgraphs with dense intra-connections and manageable inter-subgraph edges.
  - Explain the role of 1-hop neighborhood expansions more explicitly. Formally describe how overlapping regions help transmit global context signals. Consider including a short mathematical formalization.
  - Clarify how random walk structural embeddings (RWSE) and other positional encodings act as a global signal. For instance, show that the chosen positional embedding scheme can be viewed as a function  that ensures global structural information is implicitly encoded. Include a brief theoretical or intuitive argument that nodes distant in the graph have distinguishable positional encodings, thus conveying global structure to the local subgraph-level encoders.

- Empirical validation. Present at least one experiment or analysis (could be in supplementary materials) demonstrating that the chosen approaches (partitioning + expansion + positional encoding) maintain or improve performance in scenarios requiring long-range connectivity reasoning. For example, provide results on datasets known for non-local patterns and analyze whether performance degrades if the positional encoding is removed.

2) Include detailed clarification on:
 - The choice of single vs. multiple context subgraphs.
- The hyperbolic embeddings and latent codes.

3) Perform a thorough proofread to eliminate minor grammatical and typographical errors.

4) Provide an access to the source code for reproducibility check purposes.

**Audience:**

Yes, the  paper makes a  progress in the area of self-supervised learning on graphs and the design choices in this work could inspire future efforts.

**Claims And Evidence:**

Yes, pending that the careful implementation of the suggested by reviewers improvements will be made.